# Plant Biomass Conversion to Vehicle Liquid Fuel as a Path to Sustainability

Aleksandr Ketov [1,*], Natalia Sliusar [1], Anna Tsybina [1], Iurii Ketov [1], Sergei Chudinov [2], Marina Krasnovskikh [3] and Vladimir Bosnic [4]

1   Department of Environmental Protection, Perm National Research Polytechnic University, Prof. Pozdeev Str. 14, 614990 Perm, Russia
2   Bumatica Ltd., Bratskaya Str. 139, 614089 Perm, Russia
3   Department of Inorganic Chemistry, Chemical Technology and Technosphere Safety, Perm State National Research University, Bukireva Str. 15, 614068 Perm, Russia
4   Research Center RTPlast LLC, Nizhnyaya Pervomayskaya Str. 64, 105203 Moscow, Russia
*   Correspondence: alexander_ketov@mail.ru

**Abstract:** Biofuel such as linseed oil has an energy potential of 48.8 MJ/kg, which is much lower than fossil diesel fuel 57.14 MJ/kg. Existing biofuels need to increase the energy potential for use in traditional engines. Moreover, biofuel production demands cheap feedstock, for example, sawdust. The present paper shows that the technology to synthesize high-energy liquid vehicle fuels with a gross calorific value up to 53.6 MJ/kg from renewable sources of plant origin is possible. Slow pyrolysis was used to produce high-energy biofuel from sawdust and linseed oil. The proposed approach will allow not only to preserve the existing high-tech energy sources of high unit capacity based on the combustion of liquid fuels, but also to make the transition to reducing the carbon footprint and, in the future, to carbon neutrality by replacing fossil carbon of liquid hydrocarbon fuels with the carbon produced from biomass.

**Keywords:** renewable resource; energy; sustainability; carbon footprint; slow pyrolysis

## 1. Introduction

Phasing out fossil fuels to renewables is currently a global priority due to the climate change threat. Advocacy for biomass use as an energy source requires assessing the quality of biomass and the ecological impacts of bioenergy supply chains [1]. Biofuels are a renewable energy source representing an alternative to the use of fossil fuels in the transport sector. This resource is a solution to reach this target.

It is the unavoidable emissions of carbon dioxide from the oxidation of fossil fuels in internal combustion engines and diesel engines that determine the strategy of refusing to use such engines [2]. The rejection of traditional engines based on the oxidation of liquid hydrocarbons is actually a fight not against the cause but the effect, if we take into consideration the fact that liquid fuels can be produced from renewable sources of plant origin. The contribution of the transportation sector to climatic change caused by greenhouse gases can, however, be drastically reduced by the encouragement of further studies tailored toward the economical commercial production of synthetic fuels that can be used to power internal combustion engines [3].

The "carbon footprint" is not a consequence of the oxidation energy technology used, but a result of whether the fuel was obtained from a fossil or renewable source. In fact, hydrocarbon fuels obtained from plant sources do not affect the carbon balance and are carbon-neutral. The most technologically simple method for producing biofuels is the fermentation of plant biomass to produce methane. This biomethane as fuel contributes to reducing carbon dioxide emissions, gives social benefits, and increases the security supply [4]. Liquid fuels are one of the main energy carriers and predominate for vehicle

fuels, and it is also technologically possible to convert methane into other types of fuel, including liquid ones [5]. In order to generate renewable energy in a sustainable way, the energy-efficient fermentation of plant biomass seems to be a better idea. Agricultural and logging wastes appear to be the most promising sources of bioenergy and can be potentially used in the energy sector as a cheap alternative to fossil raw materials. Chemico-technological methods for converting plant and animal wastes into hydrocarbon fuels seem to be more promising, because the compact arrangement of devices having high specific productivity not only contributes to intensifying the processes of biomass conversion but also to minimizing or eliminating the impact on the environment.

Biomethane conversion to liquid fuels is not a generally accepted way, namely, liquid biofuels have some limits for use in standard engines. Biofuels produced from biomass are usually fatty acid methyl or ethyl esters from vegetable oils or animal fats. That is why biofuels always contain chemically bound oxygen, which naturally reduces the calorific value of biofuels in comparison to fossil fuels [6,7]. The oxygen content of biodiesel improves the combustion process and decreases its oxidation potential. One of the key tasks of using biofuels is to create an efficient technology for converting plant waste into energy-rich fuels. Unfortunately, in this respect, the existing types of biofuels are significantly inferior to fossil fuels. The presence of an oxygen atom in a molecule reduces the heat of combustion in accordance with the equations of Mendeleev [8].

Fossil fuels, unlike biodiesel and ethanol, are predominantly saturated, unsaturated, and aromatic hydrocarbons and do not contain oxygen heteroatoms. This structure provides not only the possibility of long-term storage of fossil fuels but also their above-mentioned high energy saturation. Furthermore, the traditional biofuel, based on vegetable oils, has a high cost and its use as a food raw material involves ethical problems [9]. Therefore, when considering the production of biofuels, special attention should be paid to non-food sources of raw materials.

Moreover, if we assume that in the process of biodegradation, the plant waste undergoes a stage of methanogenesis and enters the atmosphere in the form of methane, the gas with higher global warming potential than carbon dioxide, then the path of fresh crop residues through hydrocarbons to carbon dioxide seems to be even more promising from the point of view of climate sustainability. In this case, it is possible to focus the efforts of researchers not on the alternative energy sources, but on technologies for converting plant carbon into hydrocarbon fuels traditional for modern power engineering. Removing oxygen heteroatoms from organic substances in order to increase the calorific value of fuels is the main target in this way. Such technical solutions are known for obtaining long-carbon-chain paraffin by deoxygenating oxygen-containing groups [10].

Therefore, the aim of this research was to show the possibility of synthesizing liquid hydrocarbons with high energy values from plant materials. The objectives of the research were to provide the transformation of biomass into liquid fuel acceptable for the transportation sector. In this case, using such liquid biofuel can be considered a path to sustainability and carbon balance. In addition, in order to reduce the cost of the resulting product and dependence on food raw materials, the possibility of replacing part of the vegetable oil with vegetable waste in the form of sawdust has been considered. Based on the results obtained, it is proposed to form the concept of plant biomass conversion to vehicle liquid fuel as a path to sustainability.

## 2. Materials and Methods

The raw material used for the synthesis of biofuel was cold straight-pressed unrefined linseed vegetable oil (Russia) and sawdust from a mixture of coniferous and hardwood species typical for the taiga of the Ural Region (Russia). The sawdust had a fraction less than 1.0 mm and was dried in a drying chamber RF 115 (BINDER) in a layer of 10 mm for 24 h at 100 °C.

To obtain biofuels by transesterification of linseed oil, food alcohol was used.

It was previously shown that the slow pyrolysis of polyalkanes using the example of low-density polyethylene at high pressures and temperatures in the presence of oxygen-containing compounds leads to the formation of a mixture of predominantly oxygen-free hydrocarbons [11]. In this case, oxygen from the molecules involved in co-pyrolysis is released into the gas phase in the form of carbon dioxide.

An additional parameter capable of stabilizing the average molecular weight at a given level is maintaining the pressure in the reactor at a constant level. Thus, the slow pyrolysis of polyolefins, in particular, high-density polyethylene, under controlled pressure shows an obvious effect of the pressure in the reactor on the molecular weight of the formed hydrocarbons—an increase in pressure leads to a decrease in the average molecular weight of the resulting hydrocarbon mixture under the conditions of a flow reactor [12]. Maintaining the pressure in the reactor at a constant level leads to the removal of intermediate cracking products and stabilizes the composition of the products at an average molecular weight corresponding to the pressure in the reactor.

Based on the assumptions about the predominant participation of oxygen-containing radicals in the cracking of macromolecular compounds in the process of slow pyrolysis and the need to maintain a stable pressure in the reactor to prevent deep cracking to low molecular weight compounds, a reactor was made, schematically shown in Figure 1.

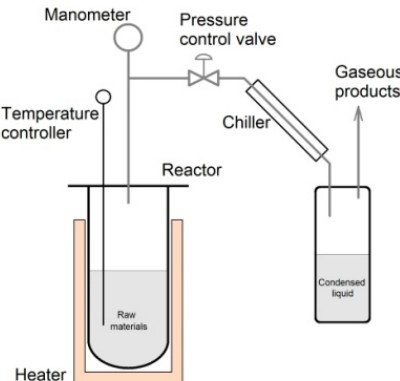

**Figure 1.** Reactor for the heat treatment of biomass.

The reactor is a steel retort designated to operate at temperatures up to 700 °C and pressures up to 12 MPa. The temperature regime of the reactor is set by an external heater. A required amount of raw material is placed in the reactor before starting its operation. 100.0 g of raw material was used in each experiment. The experiment with each sample was repeated three times. The temperature of the reaction mixture is controlled by a temperature controller, and the pressure in the reactor is controlled by a manometer.

The presence of a pressure control valve makes it possible to carry out the process of thermal destruction in two modes. In the first case, when the pressure control valve is closed, there is an uncontrolled increase in pressure in the reactor. After the pyrolysis, the reactor was cooled and the condensed product was removed from the reactor. In the second mode, the pressure control valve was set to open the valve at the pressure of 3.0 MPa. As a result, the pyrolysis gases passed after the valve through the refrigerator, where the liquid was condensed and collected in a collector, and the gases were discharged. It is assumed that in this mode, it is possible to maintain the cracking process at the level of producing predominantly medium molecular weight products $C_{13}$–$C_{25}$, removing the formed medium molecular weight products from the reaction zone.

To study the properties of biofuels, four samples were prepared in the reactor and an additional control sample by transesterification of linseed oil with vegetable ethanol according to the method described by the authors [13] at an ethanol/oil molar ratio of 12:1, NaOH (1% wt/wt), and 80 °C temperature. Sample designations and preparation conditions are given in Table 1.

**Table 1.** Conditions of obtaining biofuels in the current research.

| Sample Designation | Raw Material | Type of the Process | Pressure (MPa) | Duration (min) | Temperature (°C) |
|---|---|---|---|---|---|
| TE | Linseed oil and ethanol | transesterification | - | 90 | 80 |
| L-V | Linseed oil | pyrolysis, non-regulated pressure | Up to 9.0 | 30 | 590 |
| L-C | Linseed oil | pyrolysis, regulated pressure | 3.0 | 40 | 590 |
| LS-V | Linseed oil and sawdust (1:1 wt.) | pyrolysis, non-regulated pressure | Up to 9.0 | 30 | 590 |
| LS-C | Linseed oil and sawdust (1:1 wt.) | pyrolysis, regulated pressure | 3.0 | 40 | 590 |

The gross calorific value of the samples at a constant volume was determined using an IKA C6000 isoperibol 1/12 calorimeter (Germany).

Thermogravimetric analysis was performed with an STA 449 F1 device for synchronous thermal analysis (NETZSCH-Gerätebau GmbH, Selb, Germany), allowing the thermal analysis of a sample to be performed with a simultaneous recording of its thermal gravimetric and calorimetric characteristics. The heating rate in all the experiments was 20 degrees per minute. The thermogravimetry analysis was carried out in the inert atmosphere of argon. The flow of argon was 40 mL/min in the corresponding experiments. A platinum crucible was used for thermal analysis. The thermocouple was calibrated using reference substances. Baseline correction was performed according to the method supplied with the device. The weighed portions of the samples were measured with an accuracy of at least $\pm 1 \times 10^{-2}$ mg. The resulting data were processed with the appropriate software, NETZSCH Proteus.

## 3. Results

### 3.1. Distillation of the Product

The obtained pyrolysis products looked like dark homogeneous oils similar to fossil petroleum. Distillation takes place in a wide temperature range, similar to the distillation of natural petroleum and this process was modeled using thermogravimetry in an inert gas (Figure 2).

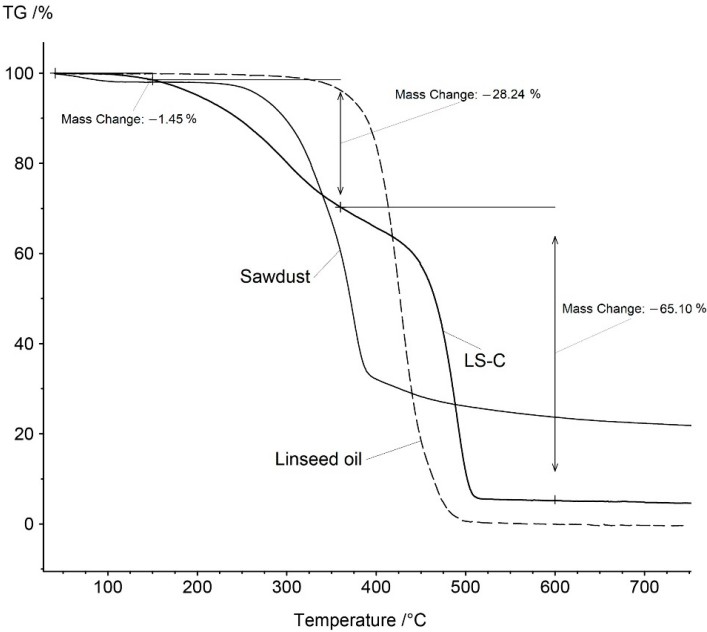

**Figure 2.** The results of the thermogravimetric analysis in an inert atmosphere for the pyrolysis product LS-C and its initial materials: sawdust and linseed oil.

The LS-C sample contains 1.45 wt.% of light hydrocarbons corresponding to the gasoline fraction, 28.24 wt.% of fraction corresponding to kerosene and diesel fuel, and 65.10 wt.% of fuel oil. Only 5.21 wt.% of the pyrolysis product stays after distillation as a pyrocarbon. The thermogravimetric curve of the pyrolysis product principally differs from the curves of initial linseed oil and sawdust. The pyrolysis product, in contrast to the initial materials, can be used for producing various liquid fuels by distillation. So, the calorific value of the pyrolysis product is the key property of the obtained liquid fuel.

### 3.2. Gross Calorific Value

The results of determining the gross calorific value are presented in Table 2.

**Table 2.** The results of determining gross calorific value.

| Sample Designation | Raw Material | Type of the Process | Gross Calorific Value (MJ/kg) |
|---|---|---|---|
| LO | Linseed oil | raw linseed oil | 48.84 |
| TE | Linseed oil and ethanol | transesterification | 45.75 |
| L-V | Linseed oil | pyrolysis, non-regulated pressure | 51.19 |
| L-C | Linseed oil | pyrolysis, regulated pressure | 53.55 |
| LS-V | Linseed oil and sawdust (1:1 wt.) | pyrolysis, non-regulated pressure | 44.99 |
| LS-C | Linseed oil and sawdust (1:1 wt.) | pyrolysis, regulated pressure | 49.06 |
| FD | | fossil diesel fuel | 57.14 |

From the presented research results, it follows that slow pyrolysis of linseed oil dramatically increases the energy value of the resulting product. The effect is explained by the removal of oxygen-containing compounds in the form of carbon dioxide into the gas phase and a corresponding increase in the gross calorific value of the liquid phase.

The addition of sawdust to the original composition as a material with a high O/C ratio makes it possible not only to obtain liquid fuel but also, under certain conditions, to obtain a product that is superior in terms of thermal properties not only to wood but also to triglyceride in the form of linseed oil.

One can think of linseed oil and sawdust as biofuels possessing low energy potential but being carbon neutral due to their origin, and fossil diesel fuel as an energy-efficient fuel but creating a high carbon footprint. The proposed technical solution for the pyrolysis of biomass under pressure makes it possible to synthesize fuels having the environmental advantages of classical biofuels devoid of the energy disadvantage of classical biofuels associated with low energy saturation.

## 4. Discussion

### 4.1. Chemical Structure and Modification of Liquid Biofuels

From the results, it is obvious that in the biofuels obtained by biomass pyrolysis, the gross calorific value increases significantly. For linseed oil, the gross calorific value increases from 48.84 MJ/kg up to 51.19 MJ/kg for conditions of non-regulated pressure and up to 53.55 MJ/kg for conditions of regulated pressure. In addition, slow pyrolysis under pressure makes it possible to involve such cheap natural raw materials as sawdust in the production of liquid fuel. Fuels obtained from a mixture of linseed oil and sawdust have a gross calorific value acceptable for liquid fuels (44.99 and 49.06 MJ/kg). Predominant elimination of oxygen-containing groups during the process of pyrolysis and transition of the resulting low-molecular-weight oxygen-containing compounds into the gas phase

may be a probable reason for this phenomenon. A prerequisite for the depletion of biofuel by oxygen-containing compounds is the implementation of the process under pressure. For the biomass with moderate oxygen content, such as vegetable oils, pressure pyrolysis results in the elimination of low molecular weight oxygen compounds and a corresponding decrease in the O/C ratio in the remaining liquid fuel. Moreover, the biofuel obtained by co-pyrolysis has a high calorific value, which permits using it for generating energy in existing engines running on high-energy fossil fuels.

The processes of reducing the O/C ratio in liquid biofuels and the corresponding increase in energy potential are described in the literature. The operational properties of any fuel directly follow its chemical structure. From this point of view, the main and common characteristic of most existing biofuels is the presence of a significant amount of compounds with heteroatoms in their composition. Thus, the study of liquid pyrolysis products resulting from the slow pyrolysis of Crambe seeds showed [14] the presence of mainly hydrocarbons 448.6 mg/g in its composition, as well as additionally nitrogenated compounds 172.2 mg/g and oxygenated compounds 37.2 mg/g. Therefore, the direct application of bio-oil as an engine fuel is often restricted by its physicochemical properties such as a low calorific value and high water, oxygen, acid, solid, and ash contents inferior to those of fossil fuels, as well as its poor thermal and storage stability.

The direct use of bio-oil in existing engines creates problems in the operation of such engines not designed for operation on such fuels. The main problems of such fuels are associated with their high viscosity and low calorific value. Therefore, the use of such fuels in existing engines requires significant modernization of engines and dilution of bio-oil with other types of fuels [15,16].

To solve these problems of improving the operational quality of fuel, various chemical and technological solutions are proposed to improve the properties of bio-oil. These are hydrotreating, hydrocracking, solvent addition, fuel blending or emulsification, esterification, supercritical treatment, catalytic cracking, and various reforming processes [17].

There is a technical possibility of separating oxygen-containing bio-oil compounds from a more energetically saturated fraction by distillation under a vacuum. Bio-oil obtained by fast pyrolysis from rice husk was distilled under vacuum [18]. As a result, the gross calorific value of the fuel almost doubled from 17.42 MJ/kg to 34.2 MJ/kg. The authors attribute this effect to a decrease in oxygen-containing compounds. In fact, the proportion of oxygen-containing compounds of formic acid and acetic acid present in the original bio-oil after vacuum distillation decreased from 7.69 and 4.56 to 0.6 and 0.36 (wt%), respectively.

The addition of bio-oil to mineral fuels currently remains the most common option for obtaining biofuels. A more promising technical solution seems to be the addition of bio-oil to mineral raw materials at the processing stage. Thus, straw-based pyrolysis oil is proposed to be added to mineral fuels at the stage of preparing raw materials in refineries [19].

In our opinion, the most promising technical solution, however, is to bring the consumer standards of biofuels, primarily the heat of combustion, to those of mineral fuels. To achieve this goal, the authors [20] propose using a hydrotreating process to improve the quality of the fuel and bring its properties closer to the mineral one. To do this, it is necessary to reduce the O/C molar ratio in bio-oil obtained by pyrolysis of the vegetable raw from 0.17–0.39 to the value not exceeding 0.005 and typical for mineral fuels.

The essence of the hydrotreating process lies, in particular, in the hydrogenation of oxygen-containing compounds. However, it is the formation of a significant amount of hydrogen that occurs in our experiments under the described slow pyrolysis conditions at temperatures above 500 °C and elevated pressure. In the described experiments, the formation of hydrogen can occur due to the steam conversion of hydrocarbons. The production of hydrogen in this way is known and described in the literature [21] as follows:

$$CH_4 + H_2O \leftrightarrow CO + 3H_2 \qquad \Delta H^\circ_{298\,K} = 206.4 \text{ kJ/mol} \qquad (1)$$

$$C_nH_{2n+2} + nH_2O \rightarrow nCO + (2n + 1)H_2 \quad \Delta H^\circ_{298\,K} > 0 \text{ kJ/mol} \qquad (2)$$



$$CO + H_2O \leftrightarrow CO_2 + H_2 \qquad \Delta H^{\circ}_{298\,K} = -41.2 \text{ kJ/mol} \tag{3}$$

$C_nH_{2n+2}$ in reaction (2) represents hydrocarbons higher than methane contained in natural gas. In total, the reactions are endothermic and proceed with an increase in the number of moles, therefore, the formation of hydrogen is facilitated by an increase in temperature and pressure.

In the case of the slow co-pyrolysis of oxygen-containing organic compounds at elevated pressures and temperatures considered in our experiments, water is always formed as a product of pyrolysis. Water, in turn, produces hydrogen in the system as a result of steam reforming. Finally, hydrogen hydrogenates oxygenated products, increasing their commercial value and reducing the O/C ratio in the resulting product. The duration of the pyrolysis process makes it possible to achieve equilibrium in the system without resorting to the use of catalysts.

On the basis of the proposed model, it is possible to explain the obtained results of increasing the gross calorific value of vegetable oil during the slow-pyrolysis at the temperature of 590 °C and pressure above 2–3 MPa. In this case, oxygen from the ether bonds of triglycerides is first split off with the formation of carbon dioxide and water. As a result of the steam reforming of hydrocarbons, free hydrogen is formed, hydrogenating, in turn, the remaining oxygen-containing compounds, vegetable oil reduces the O/C ratio and approaches, in terms of chemical composition and consumer properties, the fuels obtained from fossil raw materials. The calorific value grows for such hydrogenated fuels.

### 4.2. Carbon Footprint of Liquid Fuels

The obtained results make it possible to discuss more widely than the simple conversion of plant feedstock to engine fuel. We propose the discussion and addition of the energy and decarbonization scenarios by the transformation of solar energy into hydrocarbon engine liquid fuels through plant biomass.

In accordance with the source of origin, it is possible to conditionally divide all liquid fuels into those originating exclusively from fossil raw materials—oil, coal, or gas—into liquid fuels originating entirely from plant raw materials and a large number of intermediate options. Until the end of the 20th century, the first option had been actually universal for any type of liquid fuel, until alternative technologies began to develop, including fuel of plant origin in one form or another.

The lifecycle of plant carbon through biofuels, in comparison with the natural cycle of assimilation in the natural environment, does not contain the stage of methanogenesis and therefore can be considered more appropriate for the prevention of global warming. According to the authors of [22], methane compared with carbon dioxide, has radiative forces only 3.8 times lower, although its content in the atmosphere is 217 times lower.

The current scheme for the disposal of agricultural waste, however, leads to the path of biocarbon through the formation of methane. Indeed, it is currently in practice to add fresh crop residues to soils for agricultural purposes. In this case, the decomposition of plant residues occurs under the complex conditions of variable temperatures and oxygen concentrations, but the resulting biodegradation products are always carbon dioxide ($CO_2$), nitrous oxide ($N_2O$), and methane ($CH_4$) [23]. In the case of a dispersed distribution of degradable plant residues in the soil, it is difficult to avoid getting bio-methane into the atmosphere.

With significant emissions of methane from concentrated sources, such as landfill gas, some solutions are proposed to accelerate the oxidation of such methane by its collecting and oxidizing to generate bioenergy [24]. Another way to minimize methane emissions from landfills is to oxidize it in a bioactive cover layer [25]. Concerning the biocarbon lifecycle, the liquid fuel route appears to be more efficient in reducing methane emissions as a more dangerous greenhouse gas than carbon dioxide.

We propose the scenarios of carbon lifecycles that avoid methane emission and are based on bio-mass conversion into engine liquid fuel. Schematically, the paths through liquid fuels and natural assimilation are presented in Figure 3.

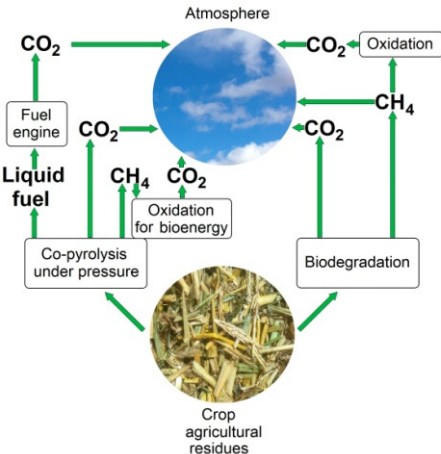

**Figure 3.** Carbon footprint of crop agricultural residues.

Under natural conditions of biodegradation of crop agricultural residues, carbon enters the atmosphere in the form of carbon dioxide in two possible ways: during the direct biodegradation of plants and after the oxidation of the resulting methane. In this case, however, a methane flow into the atmosphere necessarily remains, which is formed during biodegradation.

In the case described in this article—co-pyrolysis under pressure—there are three ways of generating carbon dioxide, and only one of them is the result of a direct co-pyrolysis process. The other two ways involve the formation of carbon-storage intermediates in the form of hydrocarbons, but in these cases, the hydrocarbons are designated for the controlled oxidation to produce bioenergy. Thus, the methane generated directly in the co-pyrolysis process is used directly in a plant for energy consumption within the process. Biofuels (the most left way on the figure) are intended to be used in engines as a replacement for fossil fuels, so this carbon enters the atmosphere only in the form of carbon dioxide.

In both ways of processing crop agricultural residues, biocarbon is released into the atmosphere in the form of carbon dioxide and can repeat its lifecycle, but in the way proposed in this article, through liquid biofuel, carbon is released into the atmosphere in the form of methane is excluded. Therefore, the proposed lifecycle of biological carbon can be considered to be more efficient in terms of the global warming problem.

*4.3. Resources for Liquid Biofuels*

From a practical point of view, the resources of plant raw materials for the production of high-energy fuels seem to be sufficient to completely replace fossil fuels.

Let us take a look at the example of Germany. In this country, the average gasoline consumption in 2020 was 452.68 thousand barrels per day [26]. Taking the average density of gasoline equal to 780 $kg/m^3$ and converting it into the SI system, we get the annual gasoline consumption of Germany at 20.5 million tons. According to the same literary source, agricultural land in Germany in 2018 was 166,450 sq.km. Considering the average yield of straw yield for different wheat species to be 3.9 tons/ha [27] and assuming that the main crop is wheat, we get the annual straw production in Germany at 64.9 million tons. It is obvious that even with significant losses during the technological conversion of straw in the process of producing biofuels and a corresponding decrease in the amount of the obtained product, the amount of liquid fuel that can be produced only from straw in Germany must be sufficient to satisfy the domestic market.

In these calculations, we have not taken into account other sources of plant raw materials in addition to wheat straw. In fact, these raw material sources are much more extensive and the use of plant raw materials to replace mineral raw materials has broad prospects. It is possible to use not only the stems of agricultural plants but also other types of waste, for example, bagasse of sunflower containing mainly cellulose (56 wt%) and protein (30 wt%) [28],

to obtain bio-oil by pyrolysis. Besides agricultural plant waste, special cultivation of various crops is possible. Moreover, working in this direction, it is possible to use aquatic plants with a high specific yield. For example, the use of various species of algae to obtain bio-oil by hydrothermal liquefaction [29] or slow pyrolysis of marine algae biomass [30] is being considered. Therefore, from a technical point of view, the use of biofuels instead of mineral ones is limited to the conversion of classical oxygen-containing biofuels into high-energy low-oxygen ones. However, this problem can be solved as shown above.

*4.4. Energy Conversion and Logistics*

Biofuel can be a source of energy only as a result of the deep oxidation of biofuel organic compounds to carbon dioxide and water. It is the reaction of deep or complete oxidation that underlies the entire thermal power industry. The opposite process of carbon dioxide and water transformation into biomass needs energy, which is consumed usually from the sun.

One cannot fail to note the fact that there are no natural processes for the direct conversion of solar energy into gaseous hydrogen, and the existing technical methods are far from perfect and necessarily include losses at each stage of energy conversion. Furthermore, to convert solar energy, devices and technological solutions are usually used, which have an ambiguous effect on environmental objects, and the issues of their recycling are still far from being resolved.

In natural conditions, to maintain the carbon cycle in balance, there is a process of photosynthesis, the opposite of combustion. If energy is released during combustion, then it is natural that the same amount of energy is required to obtain fixed carbon. As a result of oxygen photosynthesis, which appeared about 2.5 billion years ago, water splitting with the release of gaseous oxygen and carbon sequestration into organic matter due to the absorption of solar energy takes place. The authors of [31] reduce the whole process to one general equation:

$$H_2O + CO_2 + light \rightarrow C(H_2O) + O_2 \qquad (4)$$

In this case, the energy efficiency of the photosynthesis process sometimes reaches 50%.

The energy stored in biomass is solar energy in origin, and therefore the conversion of biomass to obtain any type of biofuel meets the requirements of sustainable development. If one looks at the problem from the point of view of global energy, then biofuels should be considered as an alternative solar energy storage battery, that is, meeting the requirements of sustainable development and green energy. However, the key problem with solar energy is the unevenness of its supply in time and space. It is obvious that the maximum amount of solar energy comes to the Earth during the daytime and in the equatorial regions, while the main consumers of energy are far from the equator, and energy needs change during the day. Therefore, with any technical solution for obtaining solar energy, the problems of its accumulation and transmission over considerable distances arise. An alternative way to transmit power over long distances would be to convert the power into a chemical form—for example, liquid hydrocarbon—and send the chemicals by ship [32]. The authors argue that by storing energy in the form of liquid fuels, long-distance power lines would be eliminated, and the delivered product would be storable and useful for difficult-to-electrify applications such as transport. To visualize the scale of infrastructure required, a power flow of 40 GW can be embodied by two supertankers per day full of liquid fuel. Therefore, the most effective method of accumulating and transferring solar energy appears to be its conversion by photosynthesis into biomass and further into liquid biofuel. The conducted studies show that biomass can be converted into liquid hydrocarbon directly from biomass. In this case, not only does the efficient conversion of solar energy into biomass take place, but energy is also accumulated in the form of liquid hydrocarbons having a high energy potential and convenient for transportation over long distances.

In conclusion, it should be noted that even with the transition of all the filling stations to liquid fuels identical to mineral ones in their chemical composition, it is easy to control the origin of the fuel by the presence of the $^{14}C$ isotope in biofuel. Carbon-14 ($^{14}C$) is an

ideal tracer for biogenic fuel products, due to the depletion of $^{14}$C in fossil-based fuels. Existing methods make it possible to reliably determine only 1% of the fuel of biological origin in fossil fuel [33].

## 5. Conclusions

Due to the potential depletion of fossil fuels and climate change, vehicle liquid fuel is out looking for ecologically neutral sources of energy. This leads to the huge costs of changing traditional engines and new infrastructure. However, vehicle liquid fuels can be obtained from plant biomass, becoming renewable and alternative energies to reduce the greenhouse gas emissions and other environmental impacts from the energy sector.

The novelty of the proposed study is in the proposed transformation process of obtaining high-energy liquid fuel from plant biomass using slow pyrolysis under pressure. The energy and decarbonization scenarios using the transformation of solar energy through biomass were justified. The flow bench scale reactor based on the received results is under installation and testing.

Vehicle liquid fuel with a high calorific value can be obtained from plant biomass. This fuel, in terms of energetic characteristics, is close to traditional fossil fuels, which in the future will make it possible to use it in existing power engines without technical changes.

A method for the direct reduction of the amount of oxygen-containing compounds in biofuels by slow pyrolysis at a temperature of 590 °C and elevated pressure has been studied. It was found that in the described process, the calorific value of the product obtained increases, and it becomes possible to involve biomass based on wood waste and vegetable oil for biofuel production.

The proposed liquid biofuels do not violate the existing carbon cycle and carbon balance, as they do not contain fossil carbon in their composition. The development of high-energy biofuels is in line with the reduction of carbon footprint, with the goals of sustainable development and green energy.

**Author Contributions:** Conceptualization, A.K. and N.S.; methodology, A.K.; software, M.K.; validation, V.B.; formal analysis, S.C.; investigation, A.T. and M.K.; resources, M.K.; data curation, I.K.; writing—original draft preparation, A.K.; writing—review and editing, N.S.; visualization, A.T. and I.K.; supervision, S.C.; project administration, V.B.; funding acquisition, N.S. All authors have read and agreed to the published version of the manuscript.

**Funding:** This research was funded by the Ministry of Science and Higher Education of the Russian Federation (Project No. FSNM-2020-0024 "Development of scientific basis for environmentally friendly and nature-inspired technologies and environmental management in petroleum industry").

**Institutional Review Board Statement:** Not applicable.

**Informed Consent Statement:** Not applicable.

**Conflicts of Interest:** The authors declare no conflict of interest.

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
