# Peer review of "Plant Biomass Conversion to Vehicle Liquid Fuel as a Path to Sustainability"

_resources, doi:10.3390/resources11080075_

Round 1
Reviewer 1 Report
Review comments
Article ID: resources-1812281
Title: Plant Biomass Conversion to Vehicle Liquid Fuel as a Path to Sustainability.
This article is not clear in the objective. It is unclear whether it is a research article or an outlook article. The authors mentioned that they selected materials and subjected them to a test. They produced many samples and tested them. But what is the aim of the obtained results from the tests of samples? There is no solid comparison and analysis of the tests.
However, the article must be revised, and the authors must be clear on the direction of the article. Is it research? Or is it state of a art analysis based on German fuel scenario?
Specific comments:
1. The first sentence in the abstract is a general statement. No technical problem. Delete it.
2. Line 15. This is not a technical problem. Identify a specific problem as the research problem statement.
3. If this is a research paper, what is the outcome in terms of values? You must show some results as values in the abstract.
4. After I finished reading the introduction, it looks like your technical problem is the processing of bioproducts to produce biofuels. Please mention this problem in the abstract.
5. Line 93. How do you dry the selected materials? what is the drying technique? It must be described clearly. Also, mention the tools used for the drying.
6. Line 135, table 1. Is this table for the selected materials in the current research or general. Revise the table caption to be clear.
7. Line 138. Show detailed description of the measurement procedure and the results of characterization of the materials.
8. Line 143. language problem. Revise.
9. Figure 2 is not clear. Increase the font size.
10. Line 175. which results. Be specific?
11. From line 201 to line 211. This looks like literature, not a discussion of results. In this section, you must focus on the obtained results from your measurements only. You may want to compare, OK. But don’t mention the results and comments of previous researchers.
12. Line 228 to line 230, including the equations. This is a methodology, not results and discussion. Don’t mix the discussion of results and the methods.
13. Line 250 to line 267. Again, this is not the result.
14. Is figure 3 produced from your findings, or it is general info? It isn't very clear to see such a figure in the discussion of the results.
Frankly, it is not easy to decide on the article. The authors are advised to revise and decide whether they want to direct the article towards a research article with Lab investigation and analysis or state of the art and recommendation on plant-based fuel as an alternative to fossil fuel?
Author Response
Dear Reviewer!
Thank you for your review and comments. You recommendations are very helpful and have been accepted with gratitude.
I would like to respond to your comments point-by-point.
- The first sentence in the abstract is a general statement. No technical problem. Delete it.
Accepted. The first sentence in the abstract was deleted.
- Line 15. This is not a technical problem. Identify a specific problem as the research problem statement.
Accepted. A specific problem was formulated.
- If this is a research paper, what is the outcome in terms of values? You must show some results as values in the abstract.
Accepted. Some results are shown as values in the abstract
- After I finished reading the introduction, it looks like your technical problem is the processing of bioproducts to produce biofuels. Please mention this problem in the abstract.
Accepted. The problem formulated in the abstract.
- Line 93. How do you dry the selected materials? what is the drying technique? It must be described clearly. Also, mention the tools used for the drying.
Accepted. The corresponding information was added.
- Line 135, table 1. Is this table for the selected materials in the current research or general. Revise the table caption to be clear.
Accepted. The the table caption was changed.
- Line 138. Show detailed description of the measurement procedure and the results of characterization of the materials.
Accepted. The corresponding information was added.
- Line 143. language problem. Revise.
Accepted. Mistake was revised according recommendation of Reviewer 1.
- Figure 2 is not clear. Increase the font size.
Accepted. Font sizes were increased. File “fig-2-v2.jpg” was formed.
- Line 175. which results. Be specific?
Accepted. The corresponding information was added.
- From line 201 to line 211. This looks like literature, not a discussion of results. In this section, you must focus on the obtained results from your measurements only. You may want to compare, OK. But don’t mention the results and comments of previous researchers.
Accepted. The connection added between the obtained results and literature.
- Line 228 to line 230, including the equations. This is a methodology, not results and discussion. Don’t mix the discussion of results and the methods.
Accepted. The connection added between the obtained results and literature.
- Line 250 to line 267. Again, this is not the result.
Accepted. The connection added between the obtained results and literature.
- Is figure 3 produced from your findings, or it is general info? It isn't very clear to see such a figure in the discussion of the results.
Accepted. The explanation between the Figure 3 and the results was added.
Reviewer 2 Report
I attached my recommendations in a pdf document for the authors' further consideration.

Author Response
Dear Review!
Thank you for your review and comments. You recommendations have been very helpful and accepted with gratitude.
I would like to respond to your comments point-by-point.
- Improve Abstract to give less context and more on the knowledge gap, research questions, methods and key findings.
Accepted. The abstract was improved.
- A nomenclature page, including list of abbreviations should be given
The only two abbreviations were used in the text at lines 96 and 102. These abbreviations were excluded and stayed the full names of the materials.
- In Introduction section, please outline the main aim, objectives and research questions clearly and articulate the research questions to the significance of using biomass in the transportation sector.
Accepted. The corresponding information was added to Introduction.
- Novelty of the study should be explained.
Accepted. The corresponding information was added to Conclusion.
- The authors have been discussed the previous scholars’ work in the Introduction but this is not sufficient to support the research outcomes presented in the Results section. I recommend to the authors to open-up a new section and consider these literature types as follows; systematic literature review or comprehensive literature review to study worldwide literature on using biomass as an alternative source of energy.
The article is a research type article: “The aim of this research was to show the possibility of synthesizing liquid hydrocarbons with high energy values from plant materials.” So nor systematic literature review, nor comprehensive literature review about biomass as an alternative source of energy were not included into the aims of the article. We gave enough references in order to justify our results and we would not like to transform the article into review.
- I recommend to the authors to the use this open-source software to conduct the systematic literature review on thermal comfort studies effectively. Here is the link of the open-source software tool - https://www.vosviewer.com - The authors generated the selected keywords and import the data into this software which allows the researcher to generate the visual maps. I believe that this tool could increase the scientific credibility of their research work.
We would like to thank the Review for the interesting link to such instrument as VOSviewer. The method is really important but according our opinion it is more available for reviews but not for research article.
- Materials and Methods section should be re-conceptualised, the authors should provide more detail on technical specifications of the research instruments used for the study.
Accepted. The corresponding information was added.
- The statistically representativeness of the sample size should be discussed in the Methodology. Is this sample size sufficient to make a generalisation of the study findings in energy studies? This aspect should be clarified.
Accepted. The information about samples was added.
- The method should give an honest appraisal of how the sample size were chosen and reference the work of others who have developed statistically representative sampling criteria.
Accepted. The information about samples was added.
- With regards to the identify the statistically representativeness of the sample size, please consider the statistical power of the survey. Use this open-source tool to identify the appropriate type of statistical method. Here is the link - https://www.psychologie.hhu.de/arbeitsgruppen/allgemeine-psychologie-und-arbeitspsychologie/gpower - This is the power analysis tool which helps you to identify the appropriate type of statistical method. Please use this tool and revise the statistical results presented in the Results section.
Thank you for the interesting link/ G*Power is a tool to compute statistic analyses mainly for big data. We have a small number of samples and the results are not need in statistics. From the other hand the data of instrumental methods were provided by the corresponding program software of the equipment (IKA C6000 isoperibol 1/12 calorimeter and NETZSCH Proteus).
- Results section is required additional data acquisition and interpretation of the findings. Please conduct additional experimental analysis and include them into the manuscript.
All the results have the interpretations. Explain please what “additional experimental analysis” do you mean?
- Relate your conclusions to your research questions.
Accepted. Research questions (last paragraph of Introduction) is related to Conclusions in new version.
Round 2
Reviewer 1 Report
It is highly recommended that the authors get assistance from a professional language proofreader. There are still many grammatical and spelling errors.
Also, through many rounds of re-writing, the authors will improve the article to suit the journal standard.
Reviewer 2 Report
The authors have been addressed all changes thoroughly however, the English language does not match with the expected for an academic quality. I recommend to the authors to use professional English language proof reading and editing service before the publication of their article.